# Cerebral Edema in Traumatic Brain Injury

**DOI:** 10.3390/biomedicines13071728

**Published:** 2025-07-15

**Authors:** Santiago Cardona-Collazos, Wendy D. Gonzalez, Pamela Pabon-Tsukamoto, Guo-Yi Gao, Alexander Younsi, Wellingson S. Paiva, Andres M. Rubiano

**Affiliations:** 1Meditech Foundation, Cali 760001, Colombia; cardona.santiago.meditechf@outlook.com (S.C.-C.); gonzalez.wendy.meditechf@outlook.com (W.D.G.); 2Critical Care Medicine Department, School of Medicine, Universidad del Valle, Cali 760032, Colombia; pamelapabontsukamoto@gmail.com; 3Tiantan Hospital, Capital Medical University, Beijing 100070, China; gao3@sina.com; 4Department of Neurosurgery, Heidelberg University Hospital, 69120 Heidelberg, Germany; alexander.younsi@med.uni-heidelberg.de; 5Medical Faculty, Heidelberg University, 69120 Heidelberg, Germany; 6Hospital das Clínicas, School of Medicine, Universidade de São Paulo, Sao Paulo 05508-010, Brazil; wellingson.paiva@hc.fm.usp.br; 7Global Health Research Group in Acute Brain and Spine Injuries, University of Cambridge, Cambridge CB2 0QQ, UK; 8Neuroscience Institute, Neurotrauma Group, El Bosque University, Bogotá 11001, Colombia

**Keywords:** cerebral edema, blood–brain barrier, intracranial pressure, traumatic brain injury, aquaporins, glymphatic system

## Abstract

Cerebral edema is the abnormal accumulation of fluid in any of the tissue compartments of the cerebral parenchyma. It remains a significant challenge in neurotrauma care because it contributes to secondary brain injury, affecting prognosis. This review analyzes the recent literature, including foundational studies, to describe the mechanisms of distinct types of cerebral edema following traumatic brain injury (TBI). Emerging concepts, such as the role of the glymphatic system and heme-derived inflammasomes, offer new insights into new types of edemas, differentiated by pathogenesis and potential treatments. Recent advancements in understanding these molecular mechanisms can improve therapeutic strategies, facilitating a better approach in the era of precision and personalized medicine. Although there has been notable progress, a proposal to customize treatments for diverse types of edemas is necessary to improve outcomes following traumatic brain injury. In this review, we describe the current subtypes of post-traumatic brain edemas and link them to a specific management approach.

## 1. Introduction

Cerebral edema refers to the abnormal accumulation of fluid in any of the compartments of the cerebral parenchyma [1,2,3]. It is a common response in cerebral injuries that increases the risk of secondary damage and worsens the prognosis. It represents a major contributor to poor outcomes in a variety of neurological emergencies [2,3]. Recent advances bring new insights into the molecular mechanisms of the formation and evolution of diverse types of post-traumatic cerebral edema, which show unique pathophysiological characteristics that sometimes occur concurrently. This understanding has led to the development of targeted therapeutic strategies for the management of each subtype of post-traumatic cerebral edema [1,2,3,4,5,6]. This article aims to review current pathophysiological concepts of cerebral edema formation following traumatic brain injury (TBI) and describe a new subtype of edema denominated “hemolytic edema.” It also discusses current management strategies and future perspectives that can be useful for the different health providers managing TBI globally.

## 2. Historical Aspects

Research on tissue edema dates to the late 19th century, with contributions from Ernest Starling. He proposed a model of fluid exchange through the capillary wall governed by two opposing forces: hydrostatic pressure (which pushes fluid out of the capillaries) and oncotic pressure (which draws fluid into the capillaries). Tissue edema was the result of a relative increase in hydrostatic pressure or a relative decrease in oncotic pressure, leading to fluid passage from the intravascular to the extracellular compartment [7,8].

Martin Reichardt published early works on cerebral edema in 1905. He was the first to associate cerebral fluid accumulation in brain tissue with neurological injury and distinguished between two types of edemas: “Hirnödem” (cerebral edema) and “Hirnschwellung” (cerebral swelling). This initial distinction laid the groundwork for future studies on the matter.

In 1959, Karl-Joachim Zülch introduced the terms vasogenic edema, associated with disruption of the blood–brain barrier (BBB) that allows plasma to leak into the extracellular space, and cytotoxic edema, which refers to water accumulation within glial cells and neurons due to metabolic failure, as observed in ischemic conditions [9,10].

In 1967, the neuropathologist Igor Klatzo further developed this distinction, finding that the compromised BBB in vasogenic edema allows plasma to leak into the extracellular space. He also suggested that cytotoxic edema arises from nonspecific swelling of parenchymal brain cells caused by agents that disrupt cell osmoregulation, leading to water accumulation within the cells. He elaborated on this classification, connecting it to conditions such as TBI and stroke and highlighting the importance of intracellular homeostasis in secondary neurological damage [11].

In 1989, Jean-Marc Simard, following Klatzo’s work, provided a molecular perspective by describing how the abnormal activation of ionic channels, particularly sodium, potassium, and calcium channels located in the membranes of glial cells and neurons, and the massive influx of ions contribute to cytotoxic edema during ischemic and traumatic events [11].

In 2016, Iype Cherian proposed the CSF-Shift Edema concept around the role of the Virchow–Robin spaces (VRS) in TBI [12]. New insights into the relationship between the VRS and the glymphatic system—particularly their role in clearing waste from the brain parenchyma and mediating fluid exchange between cerebrospinal fluid (CSF) and interstitial fluid—have significantly advanced our understanding of cerebral edema.

Together, these findings have laid the foundation for a contemporary framework of cerebral edema, centered around four key components: The first one is the BBB, responsible for regulating cerebral vascular permeability. The second is the VRS, which facilitates bidirectional fluid exchange between the interstitial fluid of the brain parenchyma and the CSF. The third are the ionic “pumps” and channels in the membranes of glial cells and neurons that ensure cellular osmoregulation. Finally, the fourth is the neurovascular unit, which integrates all the above, representing the interaction between the components of the BBB and neuronal activity [11,13]. The following discussion on the diverse types of post-traumatic cerebral edemas moves around these key elements.

## 3. Types of Post-Traumatic Cerebral Edema

### 3.1. Cytotoxic Edema

Cytotoxic edema results from the swelling of brain cells, particularly astrocytes, due to water and ion movement from the interstitial space to inside the cells [14]. It results from maladaptive responses of constitutively expressed and injury-induced cellular processes. These processes aim to maintain proper interstitial fluid composition but lead to abnormal ionic inflow and accumulation within cells.

The catalyst for cytotoxic edema formation is ATP depletion, which can stem from acute deprivation of oxygen and glucose or excitotoxicity after TBI [15]. The decrease in ATP disrupts the function of the Na^+^-K^+^ exchange pump and other ATP-dependent pumps, leading to the accumulation of primarily sodium and chloride within the cells [4,14]. Changes in the composition of the interstitial fluid—such as a decrease in pH from anaerobic metabolism, an increase in potassium concentration due to cell lysis, or elevations in the levels of glutamate and other excitotoxic molecules following TBI—trigger more mechanisms in astrocytes that promote the formation of cytotoxic edema [4].

The Sulfonylurea receptor 1–regulated NCCa-adenosine triphosphate (Sur1-Trpm4) channel has gained importance for its role in developing cytotoxic edema. Trpm4 is a monovalent cation channel constitutively expressed in the membrane and opens in response to intracellular calcium levels. Following a brain injury, Sur1, an ATP-binding cassette, is newly expressed and binds to Trpm4, which increases its sensitivity to calcium. This channel protects against excessive intracellular calcium concentrations, allowing sodium entry that leads to membrane depolarization, thereby limiting the driving force for calcium entry. However, in instances of more severe brain injuries, this mechanism results in excessive sodium influx, which promotes the formation of cytotoxic edema [16,17,18].

Ion influx establishes an osmotic gradient across the membrane, leading water to enter the cell through three main pathways: simple diffusion through the cell membrane, co-transport of water along with ionic transport, and, most importantly, transmembrane water channels like aquaporins [4].

Aquaporins are bidirectional water channels that function through passive transport driven by the transmembrane osmotic gradient. Aquaporins 1, 4, 9, and 11 are present in the central nervous system (CNS), with aquaporin-4 (AQP4) being the most prevalent among them. These channels are primarily located in the end feet of astrocytes and are the primary mechanism by which water enters them to neutralize the osmotic gradient responsible for cytotoxic edema (See Figure 1). Furthermore, they facilitate a high-flow pathway for the movement of water to and from the central nervous system. A comprehensive analysis of this mechanism is present in the discussion on ionic edema [19,20].

AQP4 channels facilitate osmotic water influx and are associated with adjacent ion channels and gap junctions. This connectivity plays a significant role in the dynamic regulation of brain water content. Notably, AQP4 colocalizes with inwardly rectifying potassium channels (Kir4.1) at astrocytic end feet, developing a coordinated system that couples K^+^ uptake with water transport during neuronal activity and in pathological conditions such as ischemia. This water–ion coupling exacerbates cytotoxic edema by facilitating astrocyte swelling in response to excess extracellular potassium [4,5].

Moreover, AQP4 interacts with connexin-43-containing gap junctions, which interconnect astrocytes and permit the redistribution of water and ions across the glial syncytium. While this network may help buffer localized swelling, it also has the potential to propagate edema to surrounding tissue. The dystrophin-associated protein complex (DAPC) maintains the spatial organization of AQP4 and its associated proteins. Disruption of this complex results in impaired AQP4 polarization and function [4,5].

Importantly, since cytotoxic edema represents a reorganization of cerebral water content among fluid compartments, it does not directly cause a volumetric increase in cerebral tissue. However, the influx of ions that typically reside in the interstitial compartment into their accumulation in the intracellular compartment creates a new sodium gradient across the BBB from the intravascular to the interstitial compartment, thus setting the stage for ionic edema formation that increases the volume of cerebral tissue [4,14,21].

Since the discrepancy between energy demand and supply is the primary driver of cytotoxic edema, gray matter is more susceptible to its development than white matter. Both gray and white matter become involved, resulting in a corresponding loss of differentiation when evaluated through imaging techniques such as computed tomography (CT), where the affected areas appear hypodense. Diffusion-weighted magnetic resonance imaging (DWI) reveals a reduction in the apparent diffusion coefficient (ADC), reflecting restricted water movement in edematous cells [2,22].

According to volumetric imaging studies, cytotoxic edema formation can begin as early as 30 min after injury, peak at 24 to 72 h, and persist for an additional 24 to 48 h (approximately 5 days in total) [2,23]. Hypertonic saline therapies are useful not only for increasing the osmotic gradient toward the intravascular compartment, but mostly because they generate downregulation of AQP4 in the central nervous system following TBI [24,25,26,27,28]. Because of this reason, it is recommended that the treatment for this type of edema needs to start as soon as possible and will be required to be sustained for a period of time between 3 and 5 days following the injury in order to reduce the AQP4 channels expression. Similarly, Glibenclamide has shown promising outcomes in animal studies for addressing this type of edema through the inhibition of Sur1-Trpm4. However, a definitive advantage in clinical studies remains to be clear [26,29,30,31].

### 3.2. Ionic Edema

Ionic edema typically develops after cytotoxic edema. Since the extracellular compartment is smaller than the intracellular compartment, the influx of sodium and water into brain cells caused by cytotoxic edema decreases the sodium and water content in the interstitial compartment. This creates an osmotic gradient between the interstitial and intravascular compartments, prompting the onset of ionic edema [4,32,33,34].

The formation of ionic edema involves sodium transport across the intact BBB, driven by a transmembrane osmotic gradient that draws sodium from the intravascular to the interstitial compartment. This sodium movement generates an electrical gradient that attracts chloride transport and forms an osmotic gradient, promoting the movement of water in the same direction while preventing the passage of larger molecules such as proteins (see Figure 2) [4,32,33,34]. Since the BBB is intact, the movement of these molecules into the interstitial compartment cannot be associated with leakiness, reverse pinocytosis, loss of tight junctions, or other physical processes that would permit larger molecules to cross [34].

The process takes place through sodium conduction channels located on the luminal side of the endothelial cells. These channels initially facilitate the movement of sodium from the intravascular compartment into the interior of the endothelial cells. Likewise, sodium channels on the abluminal membrane permit the efflux of sodium into the interstitial compartment, while chloride and water move through the same pathway via their respective channels [34,35].

Ionic edema represents the preliminary phase in a triphasic process that subsequently includes vasogenic edema and hemorrhagic conversion. This sequence of events impairs endothelial function. This dysfunction arises from various pre- and post-transcriptional alterations after acute brain injury, increasing the BBB’s permeability to larger molecules until it completely breaks down. The transition between phases and their respective durations are correlated with the severity of the injury [4,34].

Since ionic edema is related to the development of cytotoxic edema, it is logical to assume that treatment strategies should focus on the prompt and effective management of cytotoxic edema using the previously mentioned approaches. It is important to note that both AQP4 and Sur1-Trpm4 play roles in this type of edema, thereby reinforcing the recommendations for the previous mentioned therapies.

### 3.3. Vasogenic Edema

Vasogenic edema occurs when the BBB breaks down, allowing macromolecules and water to leak into the interstitial compartment. In TBI, this process is triggered by direct capillary damage as part of the primary insult or later in the course (after one week of the TBI) due to inflammatory changes that trigger pre- and post-transcriptional adaptations, resulting in endothelial alterations leading to BBB dysfunction, around the second week after the injury and specially surrounding brain contusions [22,35].

This type of edema develops as an ultrafiltrate of blood containing water and macromolecules but lacking in blood cells. It resembles plasma and exits the intravascular compartment through paracellular transport. Among the endothelial changes that contribute to vasogenic edema are the upregulation of matrix metalloproteinases (MMPs), which degrade the endothelial basement membrane, and vascular endothelial growth factor (VEGF), which damages the tight junction proteins of endothelial cells [2,4,10,15,34,36].

Additionally, endothelial cells can undergo actin-mediated cell retraction, which causes them to take on a rounded shape and increases their permeability. Thrombin can trigger this process, particularly in areas affected by intracerebral hemorrhage. Additional factors, such as angiopoietin and nitric oxide, contribute to endothelial permeability. The degradation of matrix proteins, along with these other mechanisms that lead to heightened endothelial permeability, contributes to the formation of vasogenic edema.

The breakdown of the BBB establishes a communication pore between the intravascular and interstitial compartments, making hydrostatic pressure the primary driver of vasogenic edema. Factors such as systemic arterial pressure, cerebral tissue pressure, and cerebral vascular resistance are among the determinants of hydrostatic pressure to consider and are, therefore, critical therapeutic targets for this type of edema [4,10,37]. In cases of TBI, vasospasm and vascular occlusions (e.g., venous thrombosis) may impact cerebral vascular resistance. Conversely, the osmotic gradient, influenced by sodium and protein concentrations, plays a minor role in vasogenic edema [4].

On CT imaging, this type of edema appears as a hypodense lesion that primarily affects white matter and features finger-like extensions while initially maintaining gray-white matter differentiation (see Figure 3). On magnetic resonance imaging (MRI), the accumulation of extracellular fluid characteristic of vasogenic edema manifests as decreased signal intensity on T1-weighted sequences and increased signal intensity on T2-weighted images. These changes are associated with hyperintense signals on Fluid-Attenuated Inversion Recovery (FLAIR) imaging and reduced fractional anisotropy on diffusion tensor imaging (DTI), reflecting disruption of normal white matter microstructure. These signal alterations occur in the absence of diffusion restrictions, distinguishing vasogenic from cytotoxic edema. The white matter, owing to its lower cellular density and higher water content, displays disrupted anisotropy, with imaging often revealing multiple disorganized and non-contiguous parallel axonal tracts, indicative of microstructural disintegration [15].

Current treatment strategies for vasogenic edema primarily involve optimizing cerebral circulatory parameters and using corticosteroids. Optimizing cerebral circulatory parameters includes maintaining systemic blood pressure at levels that ensure adequate cerebral perfusion without being excessive enough to encourage edema formation [4]. It also involves preventing and resolving vascular occlusions whenever possible. On the other hand, corticosteroids, such as dexamethasone, exert pleiotropic effects on cerebral vasculature, downregulating pro-inflammatory cytokines and VEGF production while also inducing the expression of tight junction proteins, thereby addressing the communication pore between the intravascular and interstitial compartments [37]. This effect of the corticosteroids is inefficient in the early phase of the TBI during the cytotoxic edema management. AQP-4 appears to be essential for clearing vasogenic edema. Understanding this aspect is crucial for determining the optimal duration of hypertonic saline therapy. Extending hypertonic saline treatment too far into the vasogenic edema phase or initiating it too late after the edema has arisen (after the first week of the TBI) risks worsening the post-traumatic vasogenic edema condition [4,38].

The pathophysiology of vasogenic edema may favor early rather than late secondary decompressive craniectomies for managing refractory intracranial hypertension following TBI. When patients undergo surgery early in their clinical course, the procedure optimizes tissue perfusion, addressing the primary cause of cytotoxic/ionic edema, which predominates during the initial days after injury. In later stages, it is more likely that vasogenic edema has become established, where hydrostatic pressure plays a leading role, and the BBB is already permeable. In these instances, cranial decompression reduces cerebral tissue pressure, increasing the hydrostatic pressure gradient and facilitating edema efflux [4,21,39].

### 3.4. Interstitial Edema

Interstitial edema, also known as transependymal edema, occurs when CSF leaks from the cerebral ventricles into the surrounding brain tissue. This occurs in scenarios where increased pressure within the lateral ventricles damages the ependymal ventricular lining, facilitating CSF passage from the intraventricular space to the periventricular interstitial compartment due to the dynamics of CSF circulation within the ventricular system [2,3,35,36]. Because CSF escapes through a damaged ependymal lining, the resulting edema has the same composition as CSF [2,35].

This edema results from increased intraventricular pressure due to obstructive hydrocephalus [2,35]. Multiple causes can originate from this, with intraventricular hemorrhage being the most frequent etiology in TBI patients [2].

This edema appears on a cranial CT scan as ventriculomegaly and periventricular hypodensity (Figure 4). MRI is more sensitive for its identification, showing ventriculomegaly with periventricular hypointensity on T1-weighted imaging and periventricular hyperintensity on T2-weighted imaging/FLAIR [2]. In trauma, this type of edema can be present in traumatic intraventricular bleeding or IV ventricle occlusion in the posterior fossa or in patients with non-treated hydrocephalus after cranial decompression.

For its treatment, CSF flow must be restored, either by returning it to normal flow by removing the obstruction or through external ventricular shunts, endoscopic ventriculostomies, or shunts to other anatomical spaces, which normalize the intraventricular pressure and allow the resorption of the leaked interstitial fluid [2].

### 3.5. Osmotic Edema

Osmotic edema occurs when fluid shifts from the intravascular to the interstitial compartment across an intact BBB, influenced by an osmotic gradient that arises when the osmotic concentration of plasma is lower than that of the interstitial fluid. Since osmotic forces drive this type of edema, it consists solely of water [2,10].

In the context of TBI, this can occur because of hyponatremia associated with the syndrome of inappropriate antidiuretic hormone secretion or cerebral salt-wasting syndrome or because of iatrogenic measures linked to medical care. The use of hyperosmolar agents such as hypertonic saline or mannitol, especially in vasogenic edema phases, can increase the problem. To address the issue, there is a need for treating underlying systemic abnormalities [2,10].

### 3.6. CSF-Shift Edema

CSF-shift edema results from the movement of CSF from the subarachnoid cisterns to the brain’s interstitial compartment via the perivascular spaces, also referred to as Virchow–Robin spaces. This phenomenon occurs in response to elevated pressure within the brain’s cisterns, typically due to subarachnoid hemorrhage or inadequate drainage of the cerebral interstitial compartment from a compressed VRS [4,26].

Recent findings in CSF circulation studies have shown communication between CSF in the subarachnoid space and the cerebral interstitial fluid through the VRS as part of a waste product clearance system called the glymphatic system. These VRS are microscopic fluid-filled spaces between the external and middle/internal laminae of the vessels that cross the cisternae towards and outwards the brain. Its limits are the vascular endothelial cells and the feet of the astrocytes (Figure 5). Its importance lies in the fact that it appears to be a critical link in maintaining brain homeostasis since it represents the drainage point so that CSF can enter the interstitial space to sweep away metabolic waste products and then exit again to the subarachnoid space as part of the glymphatic system making all its components work as a single fluid unit [21,39,40,41,42,43]. The recent discovery of a fourth meningeal layer, the subarachnoid lymphatic-like membrane (SLIM), which compartmentalizes the subarachnoid space and allows the direct exchange of small solutes between CSF and blood, has added even more complexity to the current model of CSF dynamics, highlighting that there are still parts of the model yet to be understood [42,44].

CSF-shift edema results from dysfunction in the VRS, which may be due to increased pressure in the subarachnoid compartment linked to traumatic subarachnoid hemorrhage (tSAH) or impaired drainage of the cerebral interstitial space caused by obstruction of the VRS [21,39]. Bleeding in the subarachnoid space results in increased pressure within the affected subarachnoid compartment (such as a cistern), continuing to rise until it matches the mean arterial pressure. At this stage, the bleeding ceases; however, the elevated pressure within the subarachnoid compartment quickly disseminates throughout the entire fluid system. This results in alterations to the transmural pressure gradients between the blood vessels, the perivascular space, and the interstitial space. This, in turn, leads to a blockage of the system with the subsequent accumulation of waste products, as well as the passage of water from the subarachnoid space to the interstitial, especially at night (Figure 6 and Figure 7) [12,39].

The treatment proposed for this type of edema is cisternostomy, a micro-neurosurgical procedure adapted from neurovascular and skull base surgery. In this procedure, the basal cisterns are open to atmospheric pressure and then irrigated, creating a pressure gradient. This allows the interstitial fluid to make a backward shift para-vascularly from the edematous brain to the basal cisterns, where the atmospheric pressure is lower. Among the theoretical benefits of the procedure are reducing swelling and washing away harmful substances that can facilitate secondary injury or prolong the homeostatic disorder [12,21,39,40,41,45,46].

Cisternostomy has shown promising results in recent studies [28,33,34]. However, the methodological quality of these studies may cloud their weight, so better-controlled studies are necessary to better understand the extent of this intervention’s effect on treating patients with acute brain injury. The procedure can be complex, particularly when addressing a swollen brain in patients with acute brain injuries and the need for a microscope.

### 3.7. Hemolytic Edema

Hemolytic edema is a newly proposed subtype of cerebral edema, characterized by the breakdown of heme products and their deleterious effects on brain tissue. It arises when blood leaks into the interstitial compartment following disruption of the blood–brain barrier (BBB), exposing cerebral tissue to plasma constituents and blood cells. Unlike vasogenic edema, hemolytic edema is distinguished by the pro-inflammatory and neurotoxic effects of heme degradation products themselves.

Blood entering the cerebral interstitial compartment starts a multi-phase process of edema formation, primarily studied in spontaneous intracerebral hematoma (ICH). However, current evidence indicates that this process is also relevant to traumatic hematomas. It begins in the moments following hematoma formation, with a higher rate of development in the first 24 to 48 h, which can persist for three weeks or more.

Hematoma formation can harm brain tissue through mechanical effects, prompting early cytotoxic edema development. Furthermore, activating the coagulation cascade to stop bleeding at the injury site causes clot retraction, which releases proteins and ions into the surrounding tissue. This raises the oncotic pressure in the interstitium. The osmotic gradient resulting from clot retraction and cytotoxic edema causes water to move into the interstitial compartment.

In the first 24 h after injury, thrombin produced by the coagulation cascade activation triggers an inflammatory reaction that further disrupts the BBB, facilitating the formation of vasogenic edema [45]. The mechanisms through which thrombin induces BBB dysfunction are multiple: First, it activates Toll-like Receptor 4 and Nuclear Factor kB, leading to the upregulation of chemokines, cytokines, and MMPs that compromise the barrier. Additionally, it promotes the production and release of chemokines and adhesion molecules that enhance the recruitment of inflammatory cells around the hematoma and increase BBB permeability, further contributing to vasogenic edema [46,47,48].

Most erythrocytes in the interstitial compartment undergo phagocytosis by microglia and macrophages. Those that escape phagocytosis face lysis due to the complement system’s formation of Membrane Attack Complexes (MACs) [49,50]. This process leads to the release of hemoglobin, which undergoes oxidation to form methemoglobin. Methemoglobin subsequently liberates heme, which turns into free iron via the action of heme oxygenase enzymes [46,47,51].

Iron induces the production of reactive oxygen species (ROS), increases nitric oxide levels, and activates microglia. These function as mediators that promote and perpetuate a pro-inflammatory environment in the perihematomal area and activate MMP, which causes the enzymatic degradation of tight junctions and the basal lamina surrounding brain capillaries. This leads to the formation of vasogenic edema, after day three to five post-injury [49,52].

Biliverdin, another product of hemoglobin oxidation, also induces neurotoxicity, further promoting and perpetuating the pro-inflammatory environment that favors edema formation [52].

Moreover, degradation products of hemoglobin in the interstitial compartment can function as osmotically active molecules, thus also encouraging the passage of fluid into this compartment (Figure 8).

On CT imaging, hemolytic edema is identified as a hypodense area in the perihematomal region in any of the different intracerebral or meningeal spaces (Figure 9). MRI reveals plasma edema as a hyperintense lesion with well-defined margins on T2-weighted images and FLAIR sequences [46].

Current treatment strategies involve surgical evacuation of the underlying hematoma and the prevention of rebleeding, particularly in scenarios where intracranial compensatory mechanisms are exhausted and where cerebral edema would be poorly tolerated. Adjunctive pharmacological interventions are under active investigation. Deferoxamine, an iron-chelating agent, has demonstrated neuroprotective properties in preclinical studies, including attenuation of perihematomal volume expansion, reduction in white matter edema, and suppression of oxidative stress-mediated neuronal apoptosis [46].

So, in general, the different types of brain edema associated with traumatic brain injury as a cause need to be differentiated in time of appearance (early or delayed), need to be defined by the affected tissue compartment (intracellular, interstitial), and need to be defined related to the integrity of the BBB (preserved or affected). These aspects are fundamental to target the specific therapy in a personalized fashion and at the appropriate time of initiation and weaning (Table 1 and Figure 10).

The simultaneous presence or convergency of different edema subtypes after a TBI explains issues like the fatality rate of a combination of primary injuries identified in a recent analysis of the CENTER-TBI imaging cohort, where patients with tSAH, acute subdural hematomas, and brain contusions show higher mortality [53]. This specific type of combination of primary injuries is associated with a combination of three subtypes of brain edemas, including CSF Shift edema (tSAH), cytotoxic edema (subdural with midline shift), and vasogenic edema (brain contusions) that will require a timely combination of focused therapies for each one of the edema subtypes.

In the context of TBI, these various types of cerebral edema should be differentiated based on three key aspects: first, the time of onset (early vs. delayed); second, the affected tissue compartment (intracellular vs. interstitial); and third, the integrity of the BBB (preserved vs. disrupted). These distinctions are critical for guiding targeted therapies in a personalized manner, both in terms of selecting the appropriate intervention and determining the optimal timing for initiation and weaning (see also Table 1).

## 4. Conclusions

Cerebral edema is a serious complication of TBI that worsens the patient’s prognosis. Today, a more precise classification of cerebral edema and its treatment is needed for personalized approaches and the application of more accurate therapeutic strategies. Diverse types of cerebral edema exist, each one with specific mechanisms and properties. Cytotoxic edema occurs when cellular metabolic processes fail, causing water to enter brain cells. On the other hand, ionic edema arises following cytotoxic edema due to an extracellular osmotic imbalance. Vasogenic edema results from the disruption of the BBB, allowing plasma to enter brain tissue. Interstitial edema occurs when the pressure in the brain ventricles increases, causing CSF to move into the interstitial compartment. Other types of brain edema include osmotic edema, caused by an osmolar imbalance; CSF-shift edema, caused by increased pressure in the subarachnoid space after tSAH; and hemolytic edema, resulting from the accumulation of blood breakdown residues in the surrounding brain tissue.

The management of cerebral edema requires an etiology-specific therapeutic approach. Vasogenic edema typically responds to corticosteroid therapy and hemodynamic optimization. In contrast, cytotoxic edema responds to hypertonic saline or mannitol. Surgical interventions, including decompressive craniectomy or evacuation of intracranial hematomas, are useful in cases of cytotoxic or hemolytic edema associated with impaired intracranial compliance. Emerging therapeutic modalities targeting ion channel modulation and neuroinflammation are under investigation and may complement advanced neurosurgical techniques, such as cisternostomy, particularly in the context of cerebrospinal fluid (CSF)-shift edema.

In conclusion, the classification and comprehensive understanding of the distinct pathophysiological subtypes of cerebral edema are critical for the development of targeted and efficacious therapeutic strategies. Progress in elucidating the molecular and cellular mechanisms underlying edema formation has enabled the implementation of mechanism-specific interventions, thereby contributing to improved clinical outcomes in patients with acute brain injuries. Nevertheless, ongoing and future research will further optimize treatment modalities and mitigate the multifactorial morbidity associated with this complex neuropathological process.

## Figures and Tables

**Figure 1 biomedicines-13-01728-f001:**
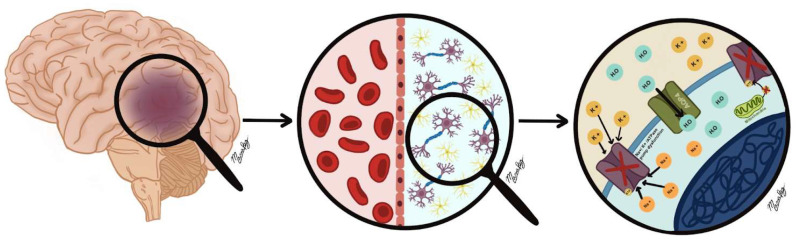
Representation of cerebral cytotoxic edema. Cytotoxic edema occurs due to the intracellular accumulation of sodium and water in astrocytes and neurons due to dysfunction of the Na^+^/K^+^-ATPase pump during ischemic events. Aquaporins, especially AQP4, facilitate water transport, exacerbating edema and contributing to cellular damage. The red crosses indicate dysfunction of the Na^+^/K^+^-ATPase pump, which prevents Na+ from leaving the cell. The black arrow within the AQP4 channel shows the passive influx of water into the cell due to cytotoxic edema.

**Figure 2 biomedicines-13-01728-f002:**
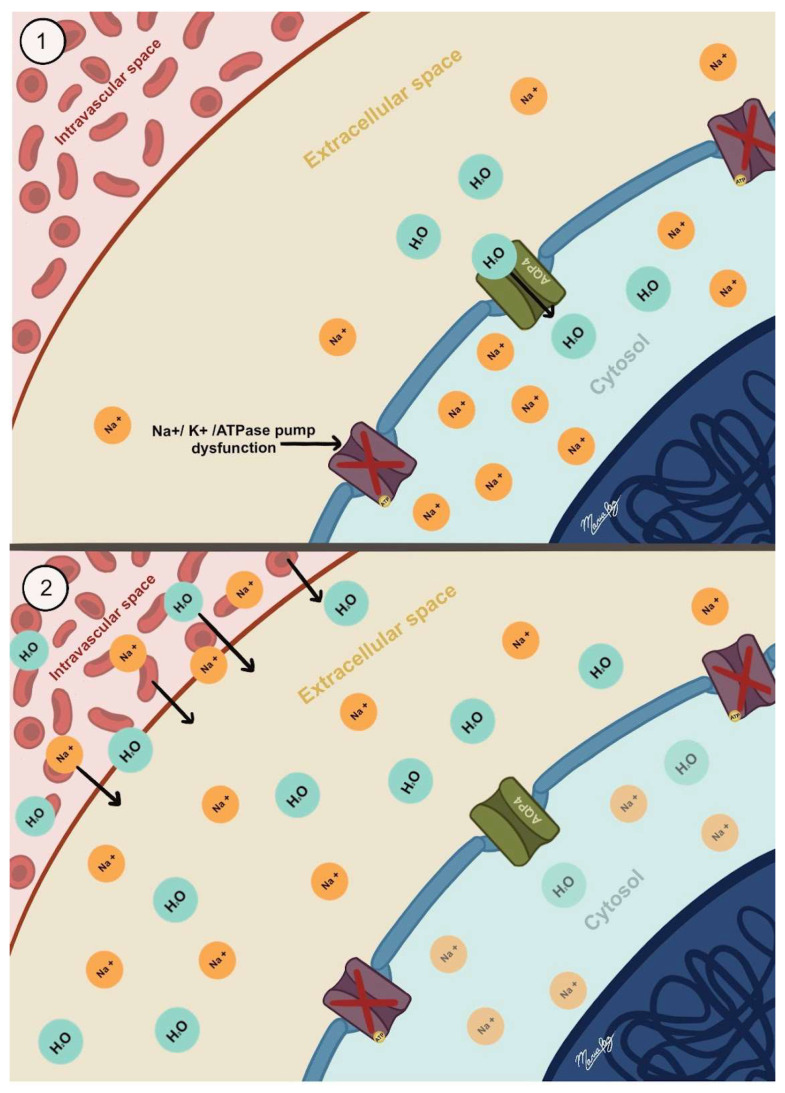
Representation of ionic edema. This type of edema arises from ionic imbalance due to cytotoxic edema, where ions accumulate inside cells. This creates an osmotic gradient that shifts fluid from the intravascular to the extracellular space across a functional BBB. ① illustrates the osmotic balance among the intravascular, extracellular, and intracellular compartments following the development of cytotoxic edema. Note the high concentration of Na^+^ and H_2_O within the intracellular space compared to the extracellular space. ② shows the subsequent movement of Na^+^ and water from the intravascular compartment into the extracellular space, triggered by the previously described imbalance, leading to the formation of ionic edema. The red crosses indicate dysfunction of the Na^+^/K^+^-ATPase pump. The black arrow within the AQP4 channel represents the passive influx of water into the cell due to cytotoxic edema. The black arrows extending from the intravascular to the extracellular space illustrate passive water movement into the extracellular compartment, as occurs in ionic edema.

**Figure 3 biomedicines-13-01728-f003:**
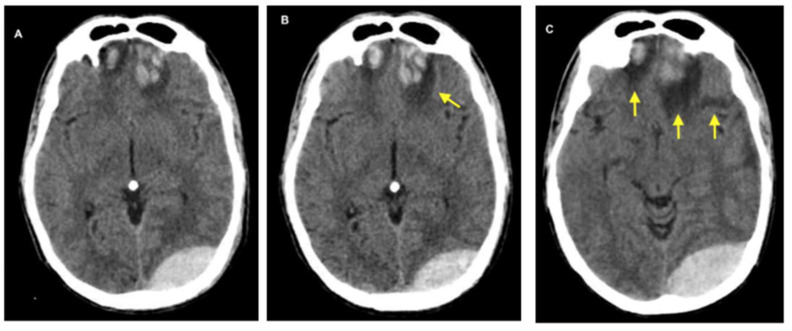
The images above show CT scans of a patient who presented with multiple frontal contusions and a left occipital epidural hematoma. The patient received conservative management, as determined by the clinical judgment of the attending neurosurgeon (who is not among the authors). The decision was based on two main factors: the patient’s Glasgow Coma Scale score consistently remained at 15, and serial imaging over the first seven days showed that the hematoma size was stable. These images illustrate the chronological evolution of the brain contusions and the progression of distinct types of associated cerebral edema. (**A**) shows early bifrontal cerebral contusions accompanied by cytotoxic edema. (**B**) displays the same contusions at a later stage, with increased surrounding edema, reflecting a combination of ongoing cytotoxic edema and the emergence of ionic, vasogenic, and hemolytic edema (described in detail below). (**C**), taken seven days post-injury, shows more pronounced edema, now predominantly vasogenic in nature. Yellow arrows highlight areas of edema in each panel.

**Figure 4 biomedicines-13-01728-f004:**
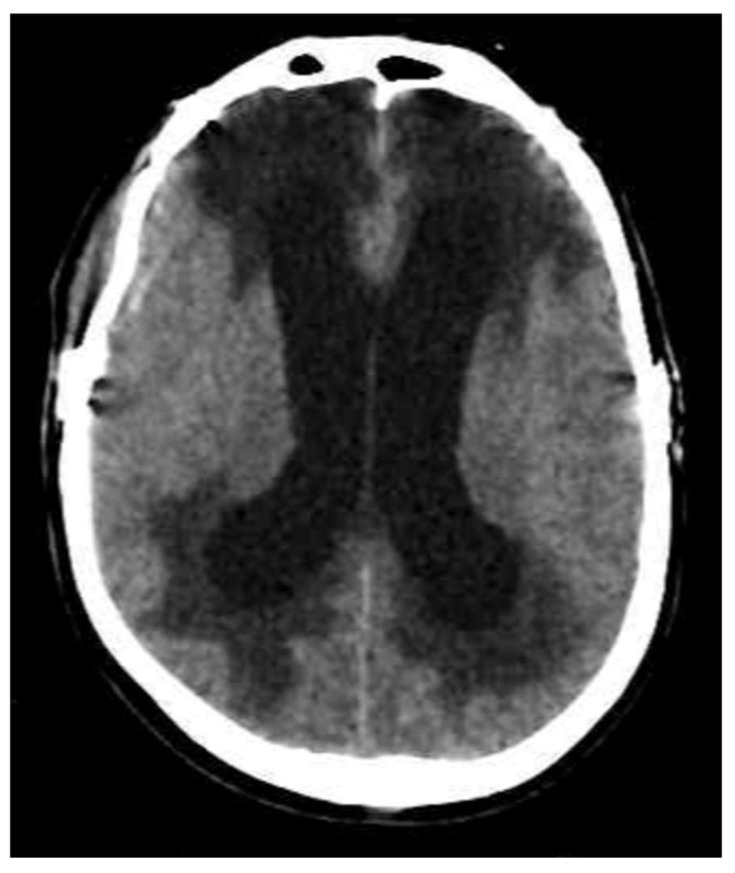
CT imaging shows a combination of ventriculomegaly and periventricular hypodensity characteristic of interstitial edema.

**Figure 5 biomedicines-13-01728-f005:**
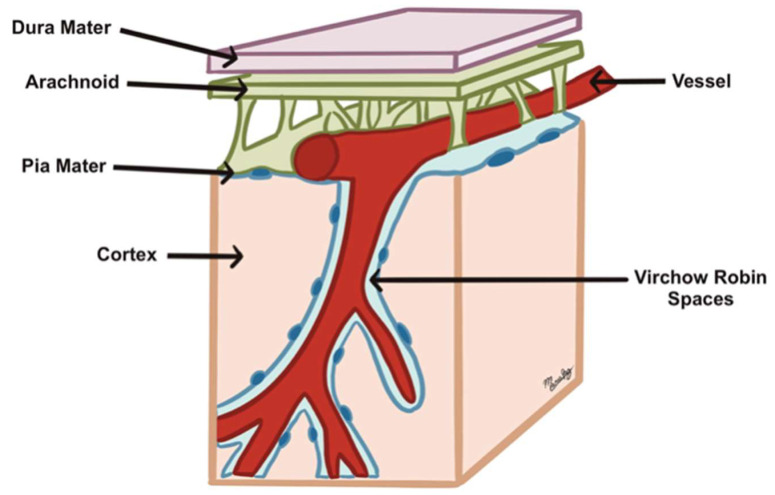
Virchow–Robin spaces: perivascular, fluid-filled canals surrounding perforating brain arteries and veins.

**Figure 6 biomedicines-13-01728-f006:**
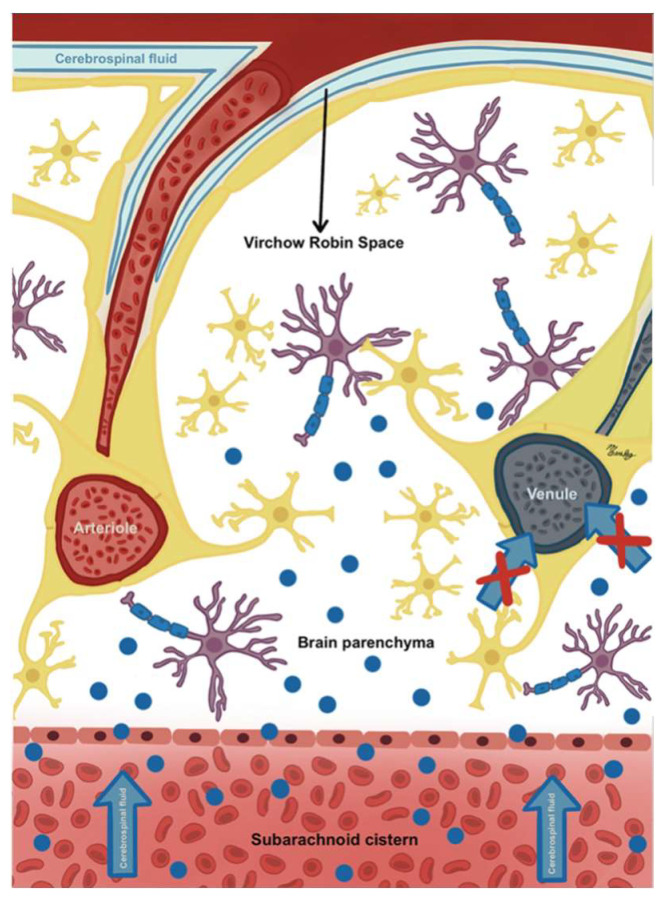
CSF-shift edema after subarachnoid hemorrhage (SAH). Blood in the subarachnoid space and basal cisterns causes a sudden rise in pressure in the subarachnoid compartment. Glymphatic impairment causes CSF to shift from the cerebral cisterns to the brain, leading to swelling. The red crosses represent the inability of interstitial fluid to drain into the venous system, a key aspect of the pathophysiology of CSF-shift edema.

**Figure 7 biomedicines-13-01728-f007:**
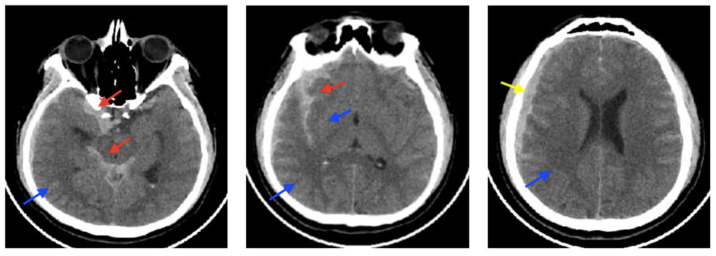
CT scan showing a thick SAH in the perimesencephalic, interhemispheric, and sylvian cisterns (red arrows), accompanied by an acute right-sided subdural hematoma (yellow arrow). Note the underlying CSF-shift edema within the adjacent white matter (blue arrows).

**Figure 8 biomedicines-13-01728-f008:**
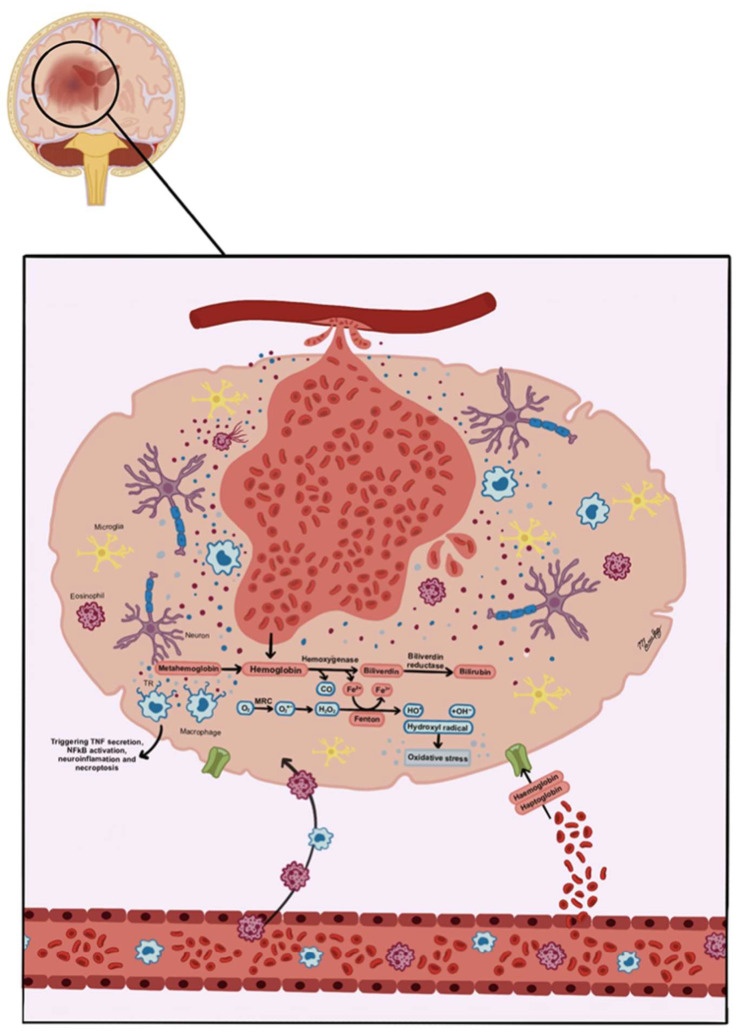
Representation of hemolytic edema pathophysiology. Around the intracerebral hematoma, in the interstitial compartment, hemolytic edema occurs (represented by blue dots) due to the degradation of hemoglobin (red dots). This process triggers different events: First, hemoglobin turns into methemoglobin, releasing iron, carbon monoxide, and biliverdin, which turns into bilirubin by biliverdin reductase. Second, the accumulation of released iron leads to oxidative stress and the formation of free radicals and reactive oxygen species (ROS). Third, the activation of microglia and macrophages stimulates the secretion of tumor necrosis factor (TNF) and the activation of NF-kB, which results in neuroinflammation and necroptosis processes.

**Figure 9 biomedicines-13-01728-f009:**
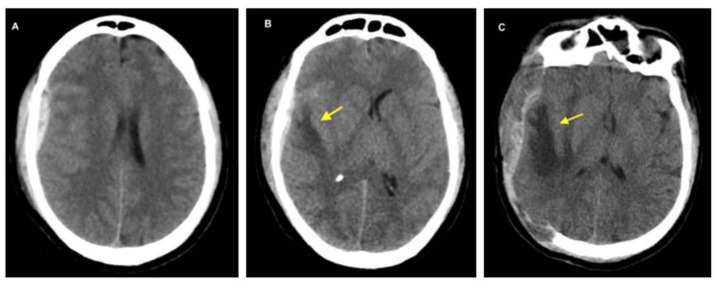
CT imaging of an acute subdural hematoma and posterior development of hemolytic cerebral edema. (**A**) demonstrates an acute subdural hematoma, few hours after injury, with a thickness of less than 10 mm and a midline deviation of less than 5 mm. (**B**) demonstrates the formation of cerebral edema in the area underlying the hematoma few days after the injury. (**C**) demonstrates the formation of more cerebral edema that required decompressive craniectomy. The yellow arrow indicates hemolytic edema.

**Figure 10 biomedicines-13-01728-f010:**
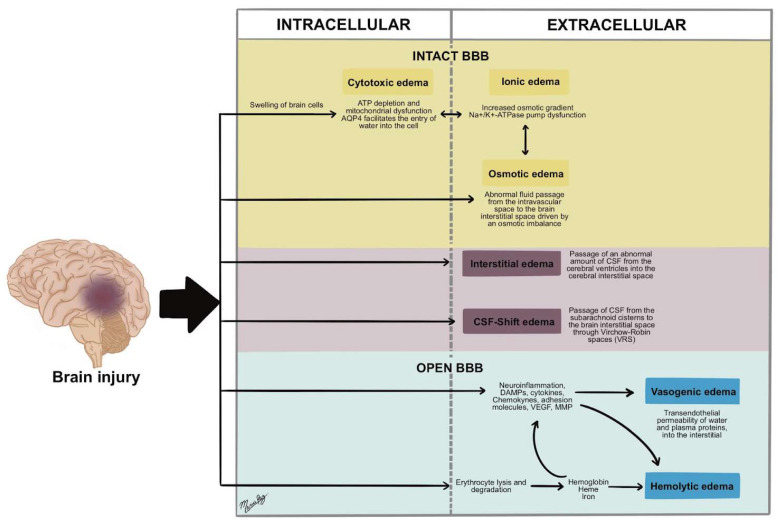
Types of cerebral edema and their distribution in various cerebral compartments.

**Table 1 biomedicines-13-01728-t001:** Types of cerebral edema and their features.

Type of Edema	BBB Integrity	Tissue Compartment	Composition	Mechanism	Proposed Treatment and Starting Time
**Cytotoxic**	Preserved	Intracellular	Water and ions	Intracellular water accumulation due to ionic imbalance from dysfunctional pumps causes fluid shift from the interstitial to the intracellular compartment.	Hypertonic saline in the first week after the injury
**Ionic**	Preserved	Interstitial	Water and ions	Fluid buildup in the interstitial compartment occurs from ionic imbalance due to cytotoxic edema, where ions accumulate inside cells. This creates an osmotic gradient that shifts fluid from the intravascular to the extracellular space across a functional BBB.	Hypertonic saline, generally starting before day 5–7.
**Vasogenic**	Compromised	Interstitial	Plasma ultrafiltrate	Fluid accumulation in the interstitial compartment caused by a dysfunctional BBB permeable to macromolecules and proteins drives a fluid shift from the intravascular to the extracellular space.	Corticosteroids, starting on the second week after injury.
**Interstitial**	Preserved	Interstitial	CSF	Abnormal fluid accumulation in the extracellular space surrounding the cerebral ventricles is caused by an increased pressure within the ventricular system that damages the ependymal lining and generates CSF passage from the intraventricular space to the periventricular extracellular space.	EVD/Ventricular shunt/Endoscopic third ventriculostomy, when identification of the imaging findings.
**Osmotic**	Preserved	Interstitial	Water and ions	An osmotic imbalance between the intravascular and extracellular spaces leads to abnormal fluid accumulation in the interstitial compartments, occurring when solute concentrations differ between the brain parenchyma and blood plasma across an intact BBB.	Achieving water homeostasis and avoiding misuse of hyperosmolar therapy, especially in the second week after injury.
**CSF-Shift**	Preserved	Interstitial	CSF/Water and ions	Fluid accumulation in the interstitial space occurs when CSF moves from the subarachnoid cisterns to the brain’s interstitial space through VRS due to increased pressure in the brain cisterns from conditions like subarachnoid hemorrhage.	Cisternostomy as solely early intervention or as coadjuvant therapy when performing a craniotomy or craniectomy for hematoma drainage or brain edema control.
**Hemolytic**	Compromised	Interstitial	Plasma ultrafiltrate	Fluid accumulation in the interstitial compartment caused by a combination of BBB disruption, inflammatory response, and osmotic gradient formed by degrading products of the heme group after intracranial hemorrhages.	Evacuation of the intracranial hemorrhage that is associated with the surrounding hemolytic process (either primary or residual bleeding)

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
