# Peer review of "Cerebral Edema in Traumatic Brain Injury"

_biomedicines, 2025, doi:10.3390/biomedicines13071728_

Round 1

Reviewer 1 Report

Comments and Suggestions for Authors

This manuscript provides a comprehensive and timely review of the mechanisms underlying brain edema in traumatic brain injury (TBI), systematically categorizing subtypes and highlighting emerging therapeutic targets. The analysis of cytotoxic, ionic, vasogenic, and hemolytic edema, coupled with discussions of the glymphatic system and Virchow-Robin spaces (VRS), enhances our understanding of fluid dynamics in TBI. However, a critical gap exists in the underrepresentation of the meningeal lymphatic system—a recently discovered component of CNS fluid clearance with significant implications for TBI-related edema. Below, I highlight key revisions needed to strengthen the manuscript:

Meningeal Lymphatic Vessels (mLVs): The Missing Link in CSF and Interstitial Fluid Homeostasis
While the review appropriately emphasizes the roles of the glymphatic system and VRS in fluid exchange, it omits mention of meningeal lymphatic vessels (mLVs), discovered in 2015, which directly drain cerebrospinal fluid (CSF), interstitial fluid, and waste products (e.g., amyloid-β) to peripheral lymph nodes. In TBI, mLV dysfunction—due to inflammation, hemorrhage, or mechanical injury—impairs CSF clearance, exacerbating interstitial and CSF-shift edema. mLV obstruction correlates with increased periventricular edema and delayed resolution of subarachnoid hemorrhage (SAH)-induced CSF transudative edema. The authors should integrate this mechanism, as mLVs complement the glymphatic system and VRS in maintaining cerebral fluid balance. Additionally, it is recommended to include content addressing therapeutic implications of targeting mLVs for post-TBI cerebral edema.

Conclusion
In summary, this review makes a valuable contribution to TBI edema research. By incorporating the meningeal lymphatic system into the fluid dynamics framework, the authors would provide a more holistic understanding of edema pathogenesis and reinforce the rationale for emerging therapies.

Author Response

Dear Reviewer,

Thank you for your thoughtful comments and for your appreciation of our manuscript.

Comment 1:

While the review appropriately emphasizes the roles of the glymphatic system and VRS in fluid exchange, it omits mention of meningeal lymphatic vessels (mLVs), discovered in 2015, which directly drain cerebrospinal fluid (CSF), interstitial fluid, and waste products (e.g., amyloid-β) to peripheral lymph nodes. In TBI, mLV dysfunction—due to inflammation, hemorrhage, or mechanical injury—impairs CSF clearance, exacerbating interstitial and CSF-shift edema. mLV obstruction correlates with increased periventricular edema and delayed resolution of subarachnoid hemorrhage (SAH)-induced CSF transudative edema. The authors should integrate this mechanism, as mLVs complement the glymphatic system and VRS in maintaining cerebral fluid balance. Additionally, it is recommended to include content addressing therapeutic implications of targeting mLVs for post-TBI cerebral edema."

Response:

We noted your interest in highlighting the role of the meningeal lymphatic vessels in the pathophysiology of cerebral edema, and we fully share your perspective on their relevance. This aspect is discussed in the section called "shift edema" in our manuscript, at least within the context of traumatic brain injury. We have included as you suggest the pathophysiology description and also the therapeutic proposed approach.

Comment 2: 

"The English could be improved to more clearly express the research."

Response: 

Regarding the suggestions for improving the English language and clarity, we will carefully revise the manuscript with the support of a native English speaker.

Once again, we sincerely appreciate your valuable input and remain attentive to any further suggestions you may have.

Sincerely,
The Authors

Reviewer 2 Report

Comments and Suggestions for Authors

The reviewed paper provides a very complex description of pathophysiology, subtypes and current management of postttraumatic brain oedema. The authors have submitted a paper with hich educational potential , particularly for younger neurosurgeons, but also  for longer time practising neurosurgeons who are used to older classifications. I appreciate the educational value of the paper, however I have two querries:

  • First, in Fig.3 if I understand the description well the large epidural occipital ( and probably also posterior fossa) haematoma has not been operated on for 7 days – if ever. In my opinion such a haematoma should be operated on as an emergency not only because of the size, but also for the risks associated with the detachment of venous sinuses from the bone which makes the treatment rathe difficult task, not talking abouthe risk of posterior fossa epidurals in general. Although the topic of the paper is different, the reason for conservative approach should be described.
  • Second , the use of corticosteroids in any type of pottraumatic brain oedema remains controversial issue. Although I personally do use short time corticosteroid treatment in patients with interstitial brain oedema not only in patients with brain tumors, but also in trauma and intracerebral haematoma cases when needed, the controversy about the use of corticosteroids soul be also noted.

Anyway it was a great pleasure for me to review this paper and after clarifying the above mentioned querries I can gladly recommed the paper for publication because of its educational value.           

Author Response

Dear Reviewer,  

Thank you for your comments and your thoughtful appreciation of our manuscript. Please find below our responses to your queries:  

Comment 1: "First, in Fig.3 if I understand the description well the large epidural occipital ( and probably also posterior fossa) haematoma has not been operated on for 7 days – if ever. In my opinion such a haematoma should be operated on as an emergency not only because of the size, but also for the risks associated with the detachment of venous sinuses from the bone which makes the treatment rathe difficult task, not talking about the risk of posterior fossa epidurals in general. Although the topic of the paper is different, the reason for conservative approach should be described."  

Response: Regarding Figure 3, the reason why this patient was selected for medical management was based on the clinical judgment of the attending neurosurgeon (who is not part of the author group). According to the medical record, at the time of evaluation, the patient showed up with a GCS of 15 and the size of the hematoma remained stable across serial imaging studies in the first 5 days. These two factors were key in the decision to opt for conservative management. We will include this explanation in the revised manuscript for greater clarity, taking into account that CT imaging and Clinical exam is not enough for defining surgical management in the new era of biomarkers and multimodal neuromonitoring in TBI patients.  

Comment 2:  "Second , the use of corticosteroids in any type of post-traumatic brain oedema remains controversial issue. Although I personally do use short time corticosteroid treatment in patients with interstitial brain oedema not only in patients with brain tumors, but also in trauma and intracerebral haematoma cases when needed, the controversy about the use of corticosteroids should be also noted."  

Response:  Regarding the use of steroids in the management of cerebral edema following traumatic brain injury, we agreed that this remains a controversial practice. However, our manuscript specifically refers to their use in the management of vasogenic edema associated with cerebral contusions, in a delayed fashion (After 1 week) which differs significantly from the use described in classical studies like the CRASH study, where steroids were used in the first 24 hours after injury.     Once again, we thank you for your valuable feedback and your input definitively will enhance the quality of the manuscript.    Sincerely, The Authors